# High-Throughput Sequencing to Identify Mutations Associated with Retinal Dystrophies

**DOI:** 10.3390/genes12081269

**Published:** 2021-08-20

**Authors:** Fei Song, Marta Owczarek-Lipska, Tim Ahmels, Marius Book, Sabine Aisenbrey, Moreno Menghini, Daniel Barthelmes, Stefan Schrader, Georg Spital, John Neidhardt

**Affiliations:** 1Human Genetics Faculty VI-School of Medicine and Health Sciences, University of Oldenburg, Ammerländer Heerstrasse 114-118, 26129 Oldenburg, Germany; fei.song@uni-oldenburg.de (F.S.); marta.owczarek-lipska@uni-oldenburg.de (M.O.-L.); 2Research Center Neurosensory Science, University of Oldenburg, 26129 Oldenburg, Germany; 3Department of Ophthalmology, Pius-Hospital, University of Oldenburg, 26121 Oldenburg, Germany; tim.ahmels@uni-oldenburg.de (T.A.); stefan.schrader@uni-oldenburg.de (S.S.); 4Eye Centre at the St. Franziskus Hospital, 48145 Münster, Germany; marius.book@augen-franziskus.de (M.B.); georg.spital@augen-franziskus.de (G.S.); 5Department of Ophthalmology, Vivantes Health Network Ltd., Neukölln Hospital, 12351 Berlin, Germany; Sabine.Aisenbrey@vivantes.de; 6Department of Ophthalmology, Ospedale Regionale di Lugano, 6900 Lugano, Switzerland; moreno.menghini@me.com; 7Department of Ophthalmology, University Hospital Zurich, University of Zurich, 8091 Zurich, Switzerland; Daniel.Barthelmes@usz.ch

**Keywords:** retinal dystrophy, mutations, novel variant, whole exome sequencing, Sanger sequencing, deep-intronic variants

## Abstract

Retinal dystrophies (RD) are clinically and genetically heterogenous disorders showing mutations in over 270 disease-associated genes. Several millions of people worldwide are affected with different types of RD. Studying the relevance of disease-associated sequence alterations will assist in understanding disorders and may lead to the development of therapeutic approaches. Here, we established a whole exome sequencing (WES) pipeline to rapidly identify disease-associated mutations in patients. Sanger sequencing was applied to identify deep-intronic variants and to verify the co-segregation of WES results within families. We analyzed 26 unrelated patients with different syndromic and non-syndromic clinical manifestations of RD. All patients underwent ophthalmic examinations. We identified nine novel disease-associated sequence variants among 37 variants identified in total. The sequence variants located to 17 different genes. Interestingly, two cases presenting with Stargardt disease carried deep-intronic variants in *ABCA4*. We have classified 21 variants as pathogenic variants, 4 as benign/likely benign variants, and 12 as variants of uncertain significance. This study highlights the importance of WES-based mutation analyses in RD patients supporting clinical decisions, broadly based genetic diagnosis and support genetic counselling. It is essential for any genetic therapy to expand the mutation spectrum, understand the genes’ function, and correlate phenotypes with genotypes.

## 1. Introduction

Inherited retinal dystrophies (RD) are characterized by progressive vision loss, often caused by degeneration of photoreceptor or retinal pigment epithelial cells. RD shows clinical variability and genetic heterogeneity. Approximately 1 in 2000 individuals worldwide are affected with different subtypes of RD, including cone-rod dystrophy (CRD), macular dystrophy (MD), retinitis pigmentosa (RP), Stargardt disease (STGD), and Usher Syndrome [1,2]. To date, more than 270 genes were identified to be associated with syndromic and non-syndromic forms of RD (https://sph.uth.edu/retnet; accessed data 30 September 2020). RD may be inherited as an autosomal dominant (AD), autosomal recessive (AR), X-linked (XL), or mitochondrial traits. Rare forms of RP have also been reported before, such as X-linked dominant inheritance, digenic diallelic, and triallelic mutations [3]. Different genes are associated with similar clinical phenotypes being obstacles to targeted genetic analyses. RD genes have diverse functions in signaling pathways, the vitamin A retinoid visual cycle, metabolism, and visual pathways [4]. In addition, mutations in the same gene can also give rise to different phenotypes [1]. Consequently, the success rates of any gene therapeutic approach depend on detailed analyses of genetic variants in each RD case.

Successful clinical trials to treat *RPE65* gene defects demonstrated the feasibility of gene replacement approaches [5,6,7,8]. The treatment was approved by the FDA (Voretigen Neparvovec, Luxturna^®^) and the EU [9,10]. Currently, genetic therapies are also being developed for other retinal diseases, e.g., targeting common pathogenetic pathways such as vascular endothelial growth factor-related pathways involved in exudative age-related macular degeneration (AMD) and diabetic retinopathy. However, in retinal dystrophies, identification and understanding of the disease-causing mutation is still challenging due to enormous genetic heterogeneity and the high variability of the clinical phenotypes [11,12,13]. Clear genotype–phenotype correlations have rarely been reported.

Next generation sequencing (NGS) is one of the most promising techniques to enable rapid identification of genetic variants in the majority of an individual’s genome. Its large potential in genetic analyses has been demonstrated in different studies aiming to detect disease-causing mutations in RD cases [11,12,14,15,16]. The number of reported disease-associated DNA variants has increased dramatically compared to the times when, e.g., Sanger sequencing or array-based mutation screening technologies provided the only sequencing platforms available. Nevertheless, Sanger sequencing is still the gold standard technique to verify disease-associated variants within larger families, as well as to screen for selected variants in genomic regions poorly covered by NGS.

Recently, several groups demonstrated the pathogenicity of the deep-intronic variants in *ABCA4*, mutations that have been associated with STGD [17,18,19,20]. Many of these deep-intronic variants affected splicing and led to, e.g., pseudoexon inclusion or exon skipping through the disruption of the splicing process [18]. Identifying and understanding disease-associated *ABCA4* alleles in non-coding regions and their correlation with clinical symptoms will help to decipher genotype-phenotype relationship. This will further be essential for the development of targeted therapeutic approaches to treat *ABCA4*-associated diseases.

In this study, we analyzed patients with clinical diagnoses of RP, CRD, STGD, MD, or Usher syndrome. We performed WES and, in selected cases, correlated the genetic results with detailed clinical analyses. We identified 37 likely causative variants in 26 unrelated patients suffering from different types of RD. We found nine novel sequence variants. Selected deep-intronic variants were analyzed in STGD cases. The complexity of our dataset confirmed the importance of detailed correlations between WES and clinical diagnoses.

## 2. Material and Methods

### 2.1. Patients and Family Members

In total, 26 index patients and 24 family members (affected and unaffected) were recruited to this study. All participants provided written informed consent and were informed in detail about the project and consequences of the study. The study adhered to the tenets of the Declaration of Helsinki and was approved by the local ethics committee (Hannover Medical School, Germany (2576-2015) and Faculty of Medicine and Health Sciences at the Carl-von-Ossietzky University Oldenburg, Germany (2018-097)).

### 2.2. Ophthalmological Investigations

Affected patients with a clinical diagnosis of either RP, MD, CRD, STGD, or Usher syndrome were included in this study. The patients underwent detailed ophthalmological examinations including fundus autofluorescence (FAF), optical coherence tomography (OCT), and electroretinogram (ERG) (Appendix A). Detailed clinical data of 10 patients and 1 further family member are shown in Appendix A.

### 2.3. DNA Extraction

EDTA blood samples from index patients and family members were collected. DNA extraction was performed using GentraPuregene Kit (QIAGEN GmbH, Hilden, Germany) according to manufacturer’s instructions.

### 2.4. Whole Exome Sequencing (WES)

We analyzed 26 patients suffering from different types of RD, among which 23 cases were analyzed by WES (15 research-based WES analyses and 8 gene diagnostic WES analyses) and 3 cases were analyzed by gene panel diagnostics. We performed the WES either as previously described [21] or by applying the Twist Human Core Exome Kit (Twist Bioscience) for DNA library preparation. Each WES analysis was enriched for exons and flanking intronic regions. WES was performed with the paired-end protocol on either the Illumina NextSeq500 or the NovaSeq6000 platform (Illumina, Berlin, Germany). Sequence reads were mapped and annotated to the human reference genome (hg19) using Varfeed and Varvis (VARVIS Version 1.15, Limbus Medical Technologies GmbH, Rostock, Germany).

### 2.5. WES Analysis Pipeline

Stepwise analysis of causative variants was performed according to standard procedures established in our lab to evaluate high-throughput datasets of cases diagnosed with various RD. An in-house gene panel containing 619 putative and known RD-associated genes was used for bioinformatic analyses to identify possible disease-causing variants.

The first steps of the bioinformatic filtering aimed to identify previously described pathogenic/likely pathogenic variants (Figure 1, Step 1). Identified disease-associated variants were re-evaluated based on available genotype–phenotype data, inheritance patterns, and publicly available allele frequencies in different populations/ethnicities (Gnom AD/GnomTotal). We considered a case to be solved when pathogenic or likely pathogenic variants were identified in this step of the analysis. In those cases where we did not find a plausible variant during the first filtering steps, we applied the in-house RD-associated gene panel including 619 candidates and disease-associated genes for further analysis (Figure 1, Step 2). In order to bioinformatically enrich for pathogenic variants in the RD panel, we applied the following filtering steps: (i) synonymous changes (not predicted to change the amino acid sequence) were excluded to enrich for more likely pathogenic variants. (ii) The gnomAD (GnomTotal) database (https://gnomad.broadinstitute.org, accessed data 25 February 2021) was used to filter all variants with allele frequencies of <0.001 (different populations and ethnicities) to select for rare variants identified in the index patient. (iii) The Varvis-internal network database “AllexesFound” (comparing the frequencies of variants detected among labs using VarVis) was set to exclude those variants that were detected ≥100 times among the contributing laboratories. If these filtering steps did not reveal plausible candidate variants, we applied the same settings to the complete WES data set (Figure 1, Step 3). Additionally, the inheritance pattern within the family was used to filter variants, e.g., excluding homozygous variants in cases with autosomal dominantly inherited RD. Occasionally, filtering steps were adapted to detect additional variants that were filtered out before. In these cases, we set the filters “GnomTotal” and “AllexesFound” to higher values to also consider more frequent variants and/or set the filter Reads-Index (Coverage of the sequenced region) to a minimum of 10 (Figure 1, Step 4). Synonymous variants were only analyzed in rare occasions.

### 2.6. Classification of Sequence Variants

We classified the identified sequence variants into 5 categories to determine their pathogenicity according to ACMG (American College of Medical Genetics). We evaluated allele frequencies (GnomAD), in silico predictions, co-segregation analysis, and published data (e.g., previously described sequence variants or descriptions on the function of gene/gene product or the consequences of mutation of the gene/gene product) [22]. We considered published evidence to conclude on the pathogenicity of each individual variant as summarized below:Pathogenic variant: A strong evidence of pathogenicity of the variant was found. This included: (i) The variant was described in literature as a clearly disease-causing mutation. (ii) Novel mutation causes damaging effects on the RNA and protein level, such as canonical splice mutation, nonsense mutation, frameshift mutation, insertion, and deletion. (iii) Mutation was validated by functional studies and segregated within the family.Likely pathogenic variant: there is evidence of pathogenicity which included: (i) splice variants were verified by in silico-predicted splice defects. (ii) The novel variant showed low allele frequencies or was predicted as pathogenic using in silico programs (e.g., SIFT, MutationTaster, PolyPhen-2, MutationAssessor). (iii) The variant was confirmed by functional studies or segregation analysis.Variant of unsure significance (VUS): There is limited evidence of pathogenicity. This variant does not fulfill the criteria of either pathogenic or benign, or the evidence is conflicting.Likely benign variant: There is evidence against pathogenicity, e.g., allele frequency is much higher than expected for the disease or in silico prediction showed conflicting results.Benign variant: There is strong evidence against pathogenicity. This variant is probably not a disease-causing mutation, because: (i) The allele frequency is higher than expected (e.g., >1% in GnomAD) for rare genetic diseases. (ii) The variant was observed in healthy populations with inheritance patterns comparable to the affected patient. (iii) Functional studies suggested no damaging effects on RNA or protein level.

### 2.7. Sanger Sequencing

Sanger sequencing was used to confirm the sequence changes detected in WES. Furthermore, previously published *ABCA4* deep-intronic variants and co-segregation analyses in family members were performed by Sanger sequencing. *ABCA4* deep-intronic variants were analyzed by Sanger sequencing using genomic DNA from those STGD patients that showed either no or only heterozygous pathogenic/likely-pathogenic variants in the coding region of the *ABCA4* gene. Sanger sequencing was performed as previously described [21]. Sanger sequencing data were analyzed with the SnapGene software (GSL Biotech LLC, Chicago, IL, USA).

## 3. Results

### 3.1. Patients and Clinical Characterizations

All patients described herein underwent ophthalmological and clinical examinations and were characterized as either RP, CRD, Usher syndrome, STGD, or MD. We analyzed 26 unrelated patients: 10 cases with RP, 5 cases with CD/CRD, 2 cases with Usher syndrome, 7 cases with STGD, and 2 cases with MD (Table 1). Detailed clinical data were available from 10 independent patients (008, 017, 019, 020, 021, 022, 023, 024, 025, 026) and one affected family member (025 II.3). Fundus and OCT images are available from seven patients (005, 008, 010, 011, 014, 018, 020) in which novel variants were detected (Appendix A). The clinical data are summarized in Appendix A. Most of the patients were Caucasians originating from Germany or Europe (Table 1). In total, 24 additional family members of 10 index patients were analyzed for familial co-segregation of variants identified in the index patient (Figure 3). Sixteen patients presented single cases where additional material from family members were not available.

### 3.2. Whole Exome Sequencing Analysis

The hg19 was used to map the WES sequence reads. High-throughput exome data of 15 affected patients (001, 002, 003, 004, 005, 006, 007, 008, 009, 010, 016, 017, 018, 019, 020) were analyzed using a bioinformatic pipeline applying the Varvis platform (VARVIS Version 1.15, Limbus Medical Technologies GmbH, Rostock, Germany). Additionally, eight patients (013, 015, 021, 022, 023, 024, 025, 026) underwent genetic diagnostic WES analyses, whereas three patients (011, 012, 014) were analyzed applying diagnostic panels. Sanger sequencing analyses verified the results from high-throughput analyses (Appendix A).

We identified 37 disease associated variants in 17 different genes, among which 9 are novel variants and 28 are previously published (Table 1). We have classified 21 variants as pathogenic variants, 4 as benign/likely benign, and 12 as variants of uncertain significance based on co-segregation analysis, type of mutation, previous publications, and bioinformatic prediction tools (Table 1). Of the novel variants, two (one in *C2orf71* and one in *MAK*) were classified to be pathogenic, whereas six (in *RIMS1*, *PDE6A*, *RP1*, *PRPF8*, *OFD1*, or *ABCA4*) were categorized as VUS (Table 1). One novel sequence alteration in *PROM1* was found to be benign (Table 1).

Disease-associated variants in *ABCA4*, *USH2A*, *RP1L1*, and *PROM1* were identified in two or more cases. *NR2E3*, *C2orf71*, *RIMS1*, *PDE6A*, *RP1*, *MAK*, *GUCA1A*, *MYO7A*, *SNRNP200*, *RPE65*, *PRPF8*, *CTNNA1*, and *OFD1* were found to carry sequence alterations in one case only (Table 1 and Figure 2A). A total of 59% of the detected variants are nonsynonymous substitution. The categories of nonsense mutations and indel mutations together account for 22% of the identified variants. In addition, 11% of the sequence alterations were predicted to interfere with splicing, likely leading to splice defect of the pre-mRNA transcript. Of note, three deep-intronic variants (8%) were identified in the *ABCA4* gene in two independent STGD-affected patients (Table 1 and Figure 2B).

### 3.3. Co-Segregation Analysis

Co-segregation analysis of sequence variants was performed in available family members. These analyses were possible in 10 cases (001, 003, 004, 005, 007, 011, 014, 022, 024, 025) (Figure 3, Appendix A).

The sequence variant c.4714C>T in *USH2A* was verified in family 003 and 004 by co-segregation analysis and found to be likely benign (Figure 3). Based on the genotype of the affected and unaffected family members, we found the c.4714C>T variant was located in a *cis* conformation with a pathogenic c.2299del variant (Figure 3). These observations were in line with previous publications [25,38]. 

The *PROM1* variant c.1069G>C was detected in the index patient with RP disease (014) and was also identified in the unaffected mother 014 I.1, indicating that this variant likely is a benign variant.

A heterozygous likely pathogenic *SNRNP200* c.3260C>T sequence alteration was found in patient 016. Interestingly, patient 016 was affected with RP as well as a duplex kidney with hypertrophy. The *SNRNP200* mutation previously was described to be associated with RP [34], but not with duplex kidney with hypertrophy. Importantly, the family anamnesis of patient 016 showed six additional family members (mother, two sisters of the mother, grandfather, granduncle, grandaunt; samples and pedigree not available) that reported kidney hypertrophy. This suggested that the *SNRNP200* c.3260C>T mutation may not only be associated with RP but also with phenotypes including duplex kidney and kidney hypertrophy. However, digenic condition cannot be excluded in this case.

Surprisingly, four *ABCA4* variants (c.5882G>A, c.6006-5T>G, c.3113C>T, c.1622T>C) were found in the index patient 022. Based on previous publications, three sequence alterations (c.5882G>A, c.3113C>T, c.1622T>C) are likely pathogenic variants [35,36,37]. Co-segregation analyses with the unaffected son 022 II.1, who carried c.5882G>A and c.6006-5T>G, suggested that both alterations locate to the same allele.

### 3.4. ABCA4 Deep-Intronic Variants Found in STDG-Patients

The seven STGD-affected patients (019, 021, 022, 023, 024, 025, 026) described herein were initially analyzed by WES analyses to search for disease-associated mutations in exons and flanking genomic regions. In the patient 019, we found a *CTNNA1* c.1310C>T variant which we classified as a VUS. All other STGD patients were associated with *ABCA4* sequence alterations.

Three of the seven STGD-affected patients (022, 025, 026) carried two or more *ABCA4* mutations which likely explain the disease phenotype (Table 1). In families 022 and 025, the co-segregation confirmed the disease association of the detected sequence variants. However, co-segregation analysis was not possible in family 026. The mutations (*ABCA4*: c.3113C>T, c.1662T>C) carried by 021 have been reported as complex allele [39]. Thus, it cannot be excluded that the detected variants locate in *cis* on the same allele and that, e.g., the presence of an additional deep-intronic variants was overlooked.

We applied Sanger sequencing analyses to verify the presence of previously published deep-intronic variants in all STGD-affected patients. We analyzed 17 previously published deep-intronic *ABCA4* variants including 16 variants in intron 30 and intron 36, and one well-characterized variant (c.4253+43G>A) in intron 28 [18,19,20,40]. Interestingly, in patient 023 two deep-intronic variants (*ABCA4*: c.4539+1770C>A and *ABCA4*: c.5196+1015A>G) were identified without any indication for an exonic sequence alteration after WES diagnostic analysis. To the best of our knowledge, the combination of the two deep-intronic variants c.4539+1770C>A and c.5196+1015A>G has not been previously reported. Our findings suggested a possible pathogenicity of the two deep-intronic *ABCA4* variants c.4539+1770C>A and c.5196+1015A>G. In contrast, 024 carried a single deep-intronic variant (*ABCA4*: c.4253+43G>A) together with the exonic frame-shifting mutation *ABCA4*: c.6601_6602delAG. The c.4253+43G>A variant was previously characterized to lead to splicing defects [19] and thus, was considered a likely pathogenic variant.

Of note, the *ABCA4* deep-intronic variant c.4540-2169A>G was frequently found in either a heterozygous (patient 019, 021, 022, 023, 026) or in a homozygous state (patient 025), suggesting a non-pathogenic nature of this sequence alteration. The allele frequency of this variant was 0.56455 according to GnomAD (https://gnomad.broadinstitute.org/, accessed data 25 February 2021).

Detailed clinic data of patients associated with STGD (including first diagnoses or differential diagnoses of STGD) were available from seven independent patients (019, 021, 022, 023, 024, 025, 026) and one affected family member (025 II.3). The clinical data are summarized in Appendix A. The OCT images of the macular region of the two STGD-affected patients 023 and 024, carrying deep-intronic variants are shown in Figure 4. Detailed clinical characterization is presented in the following.

*Patient 023:* Patient 023 has been examined in the eye clinic since he was 50 years old. Initial BCVA was OD (right eye) 20/20 and OS (left eye) 20/25 without indication of amblyopia. The patient suffered from metamorphopsia and increased photophobia. Color vision impairment was negated, family history was unremarkable. Fundus examination showed bilateral yellowish alterations of the retinal pigment epithelium (RPE) and circumscribed areas of paracentral retinal atrophy. While spared in the left eye, atrophy partly covered the fovea in the right eye. STGD could not be confirmed by standard genetic diagnostics at the time. Furthermore, there was no genetic evidence for the differential diagnoses central areolar choroidal dystrophy (CACD) and pattern dystrophy. In the course of the disease, the patient reported gradual bilateral visual deterioration, reduced contrast perception and a paracentral visual field impairment in the left eye. BCVA was reduced to 20/40 OU (both eyes), perimetry showed deep central and paracentral scotomas. Retinal imaging revealed progressive complete RPE and outer retinal atrophy (cRORA) and, consequently, regression of the former spared “foveal islands” of cones in the left eye. Multifocal electroretinogram (mERG) detected reduced amplitudes in the central area in both eyes (described in Appendix A). Color vision testing results were unspecific. In spite of the rather mild clinical appearance, STGD was still suspected.

*Patient 024:* Patient 024 has been presented at the eye clinic since he was 53 years old. Initial best corrected visual acuity (BCVA) was OD 20/20 and OS 20/25 without indication of amblyopia. The patient suffered from bilateral paracentral visual field impairment since he was 50 years old. Color vision impairment and increased photophobia were negated, the family history was unremarkable. Fundus examination showed bilateral bright spots and yellowish lipofuscin deposits. Due to their pattern in the corresponding infrared image and hyperautofluorescence, pattern dystrophy was suspected at first and genetic testing was renounced. The differential diagnosis was STGD. Over time, the patient developed progressive bilateral symptoms including metamorphopsia and visual field impairment. BCVA was moderately reduced (OD 20/25, OS 20/32), perimetry showed partly deep paracentral scotomas. Retinal imaging (OCT, fundus autofluorescence) revealed progression by detecting new, partly confluent lesions and spots of cRORA (Figure 4). Interestingly, foveal sparing could be observed. Though clinical appearance was comparably mild, STGD was suspected in the course of the disease.

## 4. Discussion

In the era of gene therapy, it is essential to develop efficient screening strategies to identify and evaluate the pathogenicity of sequence alterations in patients with genetic diseases. Here, we have shown that WES analysis can successfully be applied to cases with RD. Compared to the classic Sanger sequencing, WES shows significant advantages due to the enormous sequencing capacities. Because of the continuous improvement of NGS techniques in recent years, the quality and the coverage of the sequencing reads have increased tremendously [41]. In addition, WES technologies enable the simultaneous analysis of many patient samples on the same sequencing platform. Although whole genome sequencing has become the reality, WES is still the most reliable and cost-effective method for routine application in identifying disease-associated sequence variants in monogenetic diseases [42,43]. Since standard diagnostic testing often analyzes a subset of disease-associated genes, the modern WES analysis is a powerful alternative to overcome these drawbacks.

Nevertheless, Sanger sequencing still is the common procedure to verify sequence changes detected by WES, to analyze short disease-associated genes (e.g., GJB2 and 6) or to complement WES for difficult-to-enrich sequence regions. The genetic heterogeneity associated with many diagnoses limits the applicability of Sanger sequencing approaches, as well as of other techniques, e.g., array-based mutation screenings, arrayed primer extension (APEX) technology, or microarray-based sequencing approaches [44].

WES analyses in RD cases typically show a mutational detection rate of about 60 to 70% [12,13,45,46]. Thus, approximately one third of the cases cannot be solved by WES. Possible explanations for this observation are, among others, that (i) insertions, deletions or copy number variations are not reliably detected, (ii) synonymous mutations often are difficult to interpret, (iii) deep intronic, promotor or intragenic mutations cannot be detected, and (iv) non-genetic causes of the disease cannot be excluded [13,47]. Even in cases where in silico analyses suggested the pathogenicity of a sequence alteration, functional studies and/or co-segregation analyses might still be required to evaluate these individual sequence changes [48].

We have established a WES pipeline applying a large gene panel that included known as well as candidate genes associated with RD. In cases with an unambiguous clinical phenotype, the analysis of previously published pathogenic genes and variants is the most straightforward strategy. To minimize the possibility of overlooking potential candidate variants, analysis of the whole exome was performed.

Here, we observed consanguinity in patient 006 based on the patient data collected by clinicians, in which compound heterozygous variants (c.[3514C>A];[=], c.[130C>G];[=]) in *RP1L1* gene were identified. Interestingly, homozygous mutation c.5882G>A in *ABCA4* gene was identified in patient 026, which strongly suggests consanguinity in the family. However, the family anamnesis of the patient does not support this hypothesis. The majority of the patients (001, 002, 004, 007–011, 013–017, 019–023) are nonconsanguineous offspring in this study. In the remaining cases (003, 005, 012, 018, 024, 025), there are no clear indications of consanguinity.

In two patients diagnosed with STGD, deep-intronic *ABCA4* variants were identified. Interestingly, one of the patients even showed two deep-intronic variants, while exonic mutations were not detected. Obvious differences in the clinical presentation were not observed between the two patients carrying deep-intronic variants in *ABCA4*. The previously described disease onset of patients carrying the c.4253+43G>A variant ranged from 18 to 61 years, thus suggesting a variable age of onset or clinical severity associated with the variant c.4254+43G>A [18]. Patient 024 described herein (c.(4253+43G>A; 6601_6602delAG)) showed the first signs of STGD at an age of 50 years. In support, the intronic variant c.4253+43G>A (p.Ile1377Hisfs*3) previously was characterized and was shown to cause partial skipping of exon 28, which resulted in a frame shift and premature stop codon in exon 28 of *ABCA4* [18,40]. Our findings will help to correlate this intronic mutation with the clinical severity. Since splicing represents a highly complex nuclear process which is modulated by many, often weak, protein–protein or protein–RNA interactions, it will be interesting to further analyze the correlation between clinical presentation and haplotype-dependent efficacy of *ABCA4* exon 28 splicing.

In addition to the c.4253+43G>A sequence alteration [18,19,20,40], the two deep intronic variants c.4539+1770C>A and c.5196+1015A>G [19] might further help to explain the missing heritability in STGD-affected patients lacking clear mutations in *ABCA4* exons. Although our data led to the hypothesis that these two variants are associated with STGD, there is no experimental evidence supporting the pathogenicity of these two variants. In our study, this possible genotype-phenotype correlation was observed for the first time. Additional studies will be required to verify this notion.

In summary, WES is a reliable and cost-effective method to identify disease associated variants, which may, in selected cases, require complementary sequencing techniques to analyze deep-intronic variants. The WES pipeline described herein complements previous publications, expands the RD mutation spectrum, and broadens the phenotype–genotype correlation in the retinal diseases. A comprehensive clinical–genetical analysis of the affected patients will be essential to support the recent attempts to develop gene therapeutic approaches.

## Figures and Tables

**Figure 1 genes-12-01269-f001:**
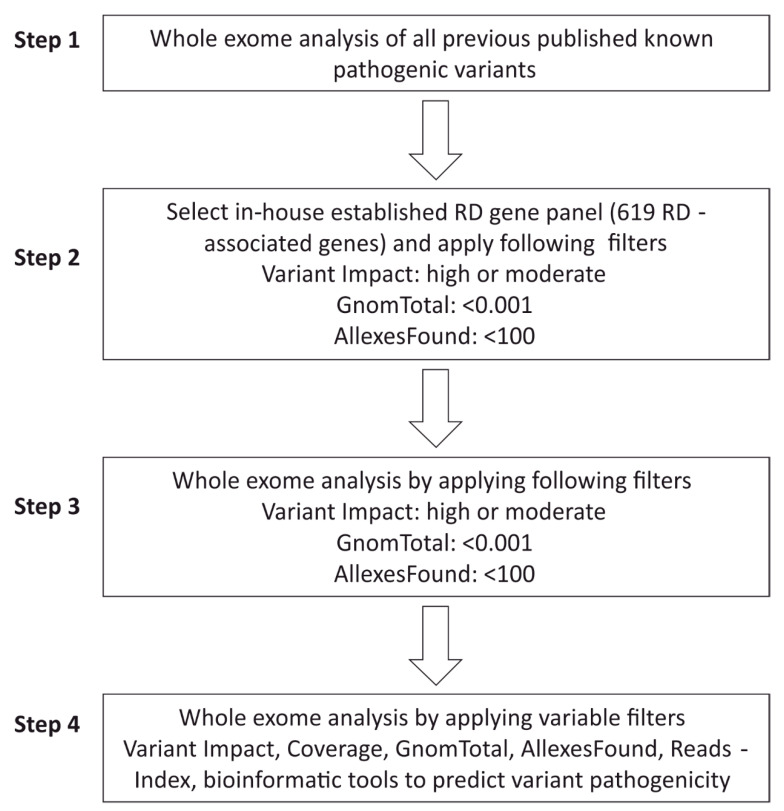
Bioinformatic pipeline to identify disease-causing mutations from WES datasets. We have performed the WES analyses in four steps with different filter settings using the Varvis platform that enabled step-by-step analyses of disease-associated variants in genomic DNAs derived from peripheral blood of individual patient. The filter “Variant Impact: high or moderate” excludes synonymous changes (not predicted to change the amino acid sequence) to enrich for more likely pathogenic variants. The filter “AllexesFound:” shows the number of persons in the allexes database that carry the variant. The filter “GnomTotal:” described the population allele frequency from merged exome and genome data sets of the gnomAD database. The filter “Coverage:” shows numbers of sequencing reads in the target region. The filter “Reads-Index:” describes the numbers of reads at the position of the variant.

**Figure 2 genes-12-01269-f002:**
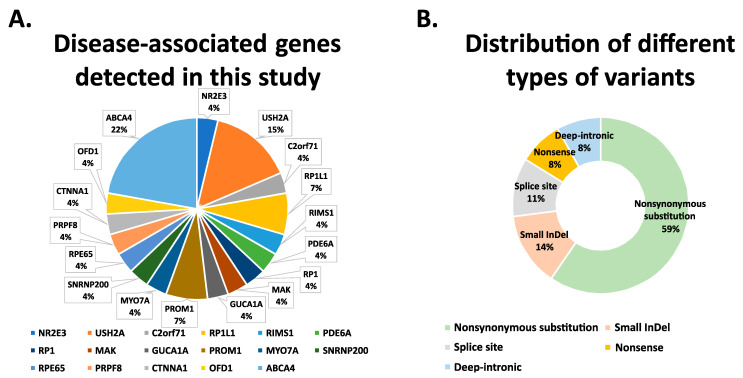
Statistical analysis of RD-associated variants detected in this study. (**A**) Disease-associated genes detected in this study. In total, 17 different genes were detected in 26 RD-affected patients. Disease associated genes are color coded. (**B**) Distribution of different types of sequence variants in known RD-associated genes. Among 37 sequenced variants, 59% nonsynonymous substitution was found, followed by 14% small InDel variants, 11% Splice variants, 8% nonsense variants, and 8% deep-intronic variants.

**Figure 3 genes-12-01269-f003:**
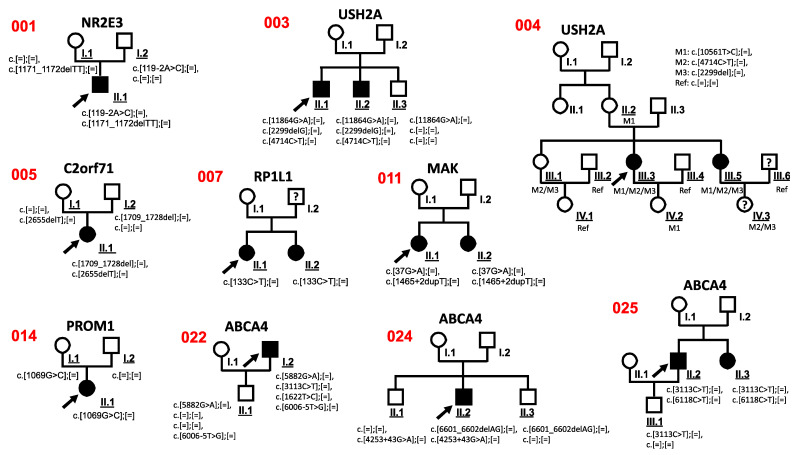
Pedigrees of all RD-affected patients for whom co-segregation analyses were performed. Black arrowheads indicate the affected index patients in each family. Filled symbols represent affected family members. Co-segregation analyses were performed in the underlined family members. The numbers on the left side of the pedigrees refer to Table 1 and Appendix A. Question mark: phenotype information not available. M: mutation. Ref: reference sequence.

**Figure 4 genes-12-01269-f004:**
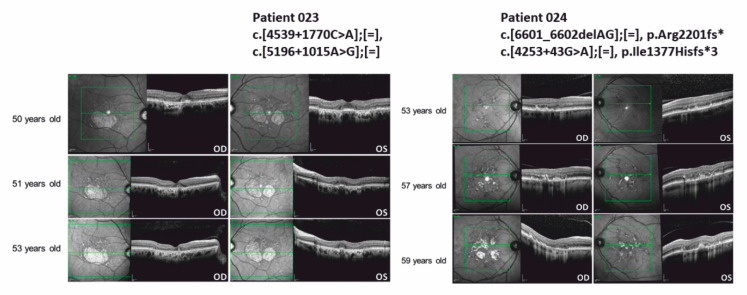
Optical coherence tomography (OCT) images of the macular region of patient 023 and 024 carrying *ABCA4* deep-intronic variants. Genetic nomenclature of the variants identified in *ABCA4* is shown above the OCT images. OD: oculus dexter, right eye. OS: oculus sinister, left eye.

**Table 1 genes-12-01269-t001:** RD-associated variants detected in this study.

Lab ID	Gender	Ancestry	Clinical Diagnosis	Disease Causing Gene	Nucleotide Change	Amino Acid Change	GnomAD Allele Frequency	Inheritance OMIM	Pathogenicity	Co-segregation (Amount of Further Family Members)	Literature, Submitter
001	m	Caucasian	RP	NR2E3 NM_014249.3	c.[119-2A>C];[=], c.[1171_1172delTT];[=]	p.?, p.Phe391Profs*15	0.0005101 Not found	AD, AR	Pathogenic Pathogenic	2	[23] Human Genetics-Radboudumc, Radboudumc
002	m	German	RP, D.D. Usher	USH2A NM_206933.2	c.[4714C>T];[=], c.[2299delG];[=]	p.Leu1572Phe, p.Glu767Serfs*21	0.0006210 0.0005376	AR	Likely Benign Pathogenic	No	[24] [25]
003	m	Romanian	Chloridemia, D.D. RP	USH2A NM_206933.2	c.[11864G>A];[=], c.[2299delG];[=], c.[4714C>T];[=]	p.Trp3955*, p.Glu767Serfs*21, p.Leu1572Phe	0.0001187 0.0005376 0.0006210	AR	Pathogenic Pathogenic Likely Benign	2	[24,25,26]
004	f	German	Usher	USH2A NM_206933.2	c.[10561T>C];[=], c.[4714C>T];[=], c.[2299del];[=]	p.Trp3521Arg, p.Leu1572Phe, p.Glu767Serfs*21	0.00006282 0.0006210 0.0005376	AR	Pathogenic Likely Benign Pathogenic	9	[27] [24] [25]
005	f	Caucasian	CRD	C2orf71 NM_001029883.2	c.[1709_1728del];[=], c.[2655delT];[=]	p.Gly570Glufs*3, p.Ser885Serfs*2	0.000004009 Not found	_	Pathogenic Pathogenic	2	[13] Novel
006	m	German	CD/CRD	RP1L1 NM_178857.6	c.[3514C>A];[=], c.[130C>G];[=]	p.Leu1172Ile, p.Pro44Ala	0.002793 0.01064	AD, AR	Likely benign Likely benign	No	[28] Illumina Clinical Services Laboratory, Illumina
007	f	German	CD	RP1L1 NM_178857.6	c.[133C>T];[=]	p.Arg45Trp	0.00002028	AD, AR	Pathogenic	1	[29]
008	m	Caucasian	CRD	RIMS1 NM_001168407.1	c.[1919C>A];[=]	p.Ser640Tyr	Not found	_	VUS	No	Novel
009	f	German	RP	PDE6A NM_000440.2	c.[607T>A];[=]	p.Phe203Ile	0.000007953	_	VUS	No	Novel
010	f	Bosnian and Herzegovinan	RP	RP1 NM_006269.1	c.[5957G>A];[=]	p.Gly1986Asp	Not found	AD, AR	VUS	No	Novel
011	f	Caucasian	RP	MAK NM_001242957.3	c.[37G>A];[=], c.[1465+2dupT];[=]	p.Gly13Ser, p.?	0.000007955 Not found	AR	Pathogenic Pathogenic	1	[30] Novel
012	m	German/Spanish	CD	GUCA1A NM_000409.4	c.[451C>T];[=]	p.Leu151Phe	Not found	AD	Pathogenic	No	[31]
013	m	German	MD	PROM1 NM_006017	c.[1117C>T];[=]	p.Arg373Cys	Not found	AD, AR	Pathogenic	No	[32]
014	f	Caucasian	RP	PROM1 NM_006017	c.[1069G>C];[=]	p.Val357Leu	Not found	AD, AR	Benign	2	Novel
015	f	German	Usher	MYO7A NM000260.4 USH2A NM_206933.2	MYO7A c.[1556G>A];[=], c.[3602G>A];[=] USH2A c.[6883G>A];[=]	MYO7A p.Gly519Asp, p.Cys1201Tyr USH2A p.Gly2295Arg	0.00001205 0.000004014 0.00002095	AD, AR; AR	Pathogenic VUS VUS	No	[33] CeGaT Praxis fuer Humangenetik Tuebingen CeGaT Praxis fuer Humangenetik Tuebingen
016	f	Caucasian	RP, duplex kidney with hypertrophy	SNRNP200 NM_014014.4	c.[3260C>T];[=]	p.Ser1087Leu	0.000003977	AD	Pathogenic	No	[34]
017	f	Caucasian	RP	RPE65 NM_000329.2	c.[1154C>T];[=]	p.Thr385Met	0.0002449	AD; AR	VUS	No	Illumina Clinical Services Laboratory, Illumina
018	f	German	RP	PRPF8, NM_006445.3	c.[1098+6del];[=]	p.?	Not found	AD	VUS	No	Novel
019	f	Ukrainian	STGD	CTNNA1 NM_001903.4	c.[1310C>T];[=]	p.(Ala437Val)	0.0005064	AD	VUS	No	Ambry Genetics
020	f	German	MD	OFD1 NM_003611.2	c.[74A>G];[=]	p.Gln25Arg	0.000005449	XLD	VUS	No	Novel
021	f	Caucasian	STGD	ABCA4 NM_000350.2	c.[3113C>T];[=], c.[1622T>C];[=]	p.Ala1038Val, p.Leu541Pro	0.001755 0.0001627	AR	Pathogenic Pathogenic	No	[35,36]
022	m	Polish/Caucasian	STGD	ABCA4 NM_000350.2	c.[5882G>A];[=], c.[3113C>T];[=], c.[1622T>C];[=], c.[6006-5T>G];[=]	p.Gly1961Glu, p.Ala1038Val, p.Leu541Pro, p.?	0.004564 0.001755 0.0001627 Not found	AR	Pathogenic Pathogenic Pathogenic VUS	1	[35,36,37] Novel
023	m	Caucasian	STGD	ABCA4 NM_000350.2	c.[4539+1770C>A];[=], c.[5196+1015A>G];[=]	p.?, p.?	Not found Not found	AR	VUS VUS	No	[19]
024	m	Caucasian	STGD	ABCA4 NM_000350.2	c.[6601_6602delAG];[=], c.[4253+43G>A];[=]	p.Arg2201fs* p.Ile1377Hisfs*3	Not found 0.004694	AR	Pathogenic Pathogenic	2	[36] [19]
025	m	German/Caucasian	STGD	ABCA4 NM_000350.2	c.[3113C>T];[=], c.[6118C>T];[=]	p.Ala1038Val, p.Arg2040*	0.001755 0.00001415	AR	Pathogenic Pathogenic	2	[35] EGL Genetic Diagnostics
026	m	Afghan	STGD	ABCA4 NM_000350.2	c.[5882G>A];[ 5882G>A], c.[123G>A];[=]	p.Gly1961Glu, p.Trp41*	0.004564 0.000003978	AR	Pathogenic Pathogenic	No	[37] EGL Genetic Diagnostics

D.D.: Differential diagnosis. RP: retinitis pigmentosa. STGD: Stargardt disease. MD: macula dystrophy. CRD: Cone-rod dystrophy. CD: cone dystrophy. Usher: Usher syndrome.

## Data Availability

The data that support the findings of this study are available on request from the corresponding author. The complete data sets are not publicly available due to privacy or ethical restrictions.

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
