# Peer review of "High-Throughput Sequencing to Identify Mutations Associated with Retinal Dystrophies"

_genes, 2021, doi:10.3390/genes12081269_

Round 1

Reviewer 1 Report

Summary: In the manuscript ‚High-throughput sequencing to identify mutations associated with retinal dystrophies’ the authors applied WES-based analysis to 26 unrelated patients. For identification of disease-related mutations, a four-step filtering was used for separation of already-known (solved) variants, comparison with an in-house RD gene panel and selection of rare variants using GnomTotal and AlexesFound. In the study clinical and sequencing data are thoroughly evaluated and presented, which provides relevant knowledge to the community.

Only some minor points should be addressed before publication:

- Sanger sequencing results (chromatogram) should be added to the supplement, not only to show prober results but also to add information of adjacent sequences.

- Figure 2.A. : The gene-Percentage panels are overlapping and hiding some information, here some re- organization is needed.

- Page 11: Formatting of text part ‘we analyzed’ in wrong type size.

Reviewer 2 Report

Song et al. uses high throughput technologies to identify novel pathogenic mutations causing retinal dystrophies. The work is impressive and will help in precision sciences. After reviewing, I recommend for a major revision. Comments are described below.

  1. I highly recommend authors to add all pedigrees Figure 3, along with reported nucleotide/ amino acid change. It will look impressive and easy to understand the inheritance pattern with the mutation.
  2. I recommend authors to add clinical pictures of fundus and OCT, of those patients in which novel mutation has been detected. It will help other researchers to understand the phenotypes.
  3. Did authors observe any consanguinity in their familiar cases. Please add this in discussion.
  4. In Table 1, Under gender column, please represent female as ‘F’ instead of w.
  5. In Table 1, Please check the spelling of allele in allele frequency column.
  6. Please add visual acuity details in Supplementary Clinical Table S1. It will give more details on association between pathogenicity of mutation and visual acuity.
  7. Please discuss about the diverse functions of RD genes in visual pathway, metabolism, signaling in the introduction. Use reference https://doi.org/10.1007/s10633-018-9654-x. 
  8. Please add line in introduction regarding rare forms such as x-linked dominant, digenic diallelic, triallelic mutations. I  recommend authors use reference from DOI:1016/j.jcjo.2018.02.008 paper.

Round 2

Reviewer 2 Report

Authors have answered all concerns. The manuscript is good for publication.
